# Three Northern Regions Shelter Forest contributed to long-term increasing trend of biogenic isoprene emissions in Northern China

**Authors:**
Xiaodong Zhang[1], Tao Huang[1*], Leiming Zhang[2], Yanjie Shen[1], Yuan Zhao[1], Hong Gao[1], Xiaoxuan Mao[1], Chenhui Jia[1], Jianmin Ma[1,3]*

**Affiliation:**
[1]Key Laboratory for Environmental Pollution Prediction and Control, Gansu Province College of Earth and Environmental Sciences, Lanzhou University, Lanzhou 730000, P. R. China
[2]Air Quality Research Division, Environment Canada, Toronto, Ontario, M3H 5T4, Canada
[3]CAS Center for Excellence in Tibetan Plateau Earth Sciences, Beijing 100101, China

**\*Corresponding author:** Jianmin Ma, Tao Huang
Tel: +86 15293166921, fax: +86-931-8911843, email: jianminma@lzu.edu.cn;
huangt@lzu.edu.cn

**Abstract**

To assess the long-term trends of isoprene emissions in Northern China and the impact of the Three Northern Regions Shelter Forest (TNRSF) on these trends, a database of historical biogenic isoprene emissions from 1982 to 2010 was developed for this region using a biogenic emission model for gases and aerosols. The total amount of the biogenic isoprene emissions during the three decades was 4.4 Tg in Northern China and 1.6 Tg in the TNRSF, with annual emissions ranged from 132,000 to 176,000 ton yr$^{-1}$ and from 45,000 to 70,000 ton yr$^{-1}$, respectively, in the two regions. Isoprene emission fluxes have increased substantially in many places of the TNRSF over the last three decades due to the growing trees and vegetation coverage, especially in the Central-North China region where the highest emission incline

reached to 58% from 1982 to 2010. Biogenic isoprene emissions produced from
anthropogenic forests tended to surpass those produced from natural forests, such as
boreal forests in Northeastern China. The estimated isoprene emissions suggest that
the TNRSF has altered the long-term emission trend in North China from a decreasing
trend during 1982 to 2010 (slope=-0.533, $R^2$=0.05) to an increasing trend for the same
period of time (slope=0.347, $R^2$=0.014), providing strong evidence for the change in
the emissions of biogenic volatile organic compounds (BVOCs) induced by the
human activities on decadal or longer time scales.
**Key words:** Volatile organic compounds, human activities, biogenic emissions,
statistical trend
**1. Introduction**
While trees and plants can efficiently remove pollutants from the atmosphere (Nowak
et al., 2006,2014; Myles et al., 2012; Camporn, 2013; Fenn et al., 2013; Adon et al.,
2013; Zhang et al., 2015), they also contribute to air pollution through atmospheric
chemistry. It has been widely acknowledged that terrestrial ecosystems release large
quantities of reactive biogenic volatile organic compounds (BVOCs) into the
atmosphere as a significant product of biosynthetic activities of trees and plants
(Purves et al., 2004; Zemankova and Brechler, 2010). BVOCs play important roles in
tropospheric chemistry, carbon budget, and global climate change (Purves et al., 2004;
Nichol and Wong, 2011; Aydin et al., 2014). For example, BVOCs are precursors of
surface ozone formation in the presence of nitrogen oxide ($NO_x$) (Penuelas et al.,
2009; Penuelas and Staudt, 2010). It has been shown that VOC emissions from
biogenic sources have far exceeded those from anthropogenic sources (Guenther et al.,
1994, 1995; Aydin et al., 2014).
Among the three dominant VOCs (isoprene, monoterpenes, oxygenated
compounds) contributing to BVOC emission fluxes, isoprene accounts for 70% of the
total BVOC emissions globally (Guenther et al., 2006; Helmig et al., 2013; Aydin et
al., 2014) and about 50% in China (Song et al., 2012, Li et al., 2013). In particular,
terrestrial plant foliage is thought to be the major source of atmospheric isoprene
which releases over 90% of isoprene from global forests (Lamb et al., 1987; Guenther
et al., 2006). Extensive investigations have been conducted over the past several
decades to assess BVOC emissions and their potential influences on tropospheric
chemistry and carbon cycle (Lamb et al., 1987; Ceron et al., 2006; Muller et al., 2008;
Chang et al., 2009; Pacifico et al., 2009; Zemankova and Brechler, 2010; Guo et al.,
2013; Calfapietra et al., 2013). Efforts have been also made to measure and simulate
BVOC emissions in China (Wei et al., 2007; Chen et al., 2009; Song et al., 2012; Li et
al., 2013). A recent study by Song et al. (2012) revealed that the annual BVOC
emission in Eastern China was $11.3 \times 10^6$ t, of which 44.9% was isoprene, followed by
monoterpenes at 31.5%, and other VOCs at 23.6%. The study also showed high
isoprene emissions in boreal forests in Northeastern China, on Qinling – Ta-Pa
Mountains in central China, and in Southern China. Li et al. (2013) estimated the
2003 China's total BVOC emission as 42.5Tg, of which 55% was isoprene emission.
BVOC emissions are often thought to be static on decadal or longer time scales
because forest coverage from regional to global scales is assumed to be at steady state
(Sanderson et al., 2003; Purves et al., 2004). However, there are concerns for the
potential impacts of climate change and changes in underlying vegetation coverage on
isoprene emissions because leaf level emission intensity depends on biological and
meteorological conditions (Turner et al., 1991; Constable et al., 1999; Ashworth et al.,
2010; Arneth et al., 2008, 2011). Several modeling studies were conducted to assess
the interactions between biogenic isoprene emissions and climate change as well as
the human activities (Constable et al., 1999; Sanderson et al., 2003). Using the USDA
(the United States Department of Agriculture) Forest Service Inventory Analysis
(FIA), Purves et al (2004) estimated decadal changes in BVOC emissions in the
Eastern US between the 1980s and 1990s caused by changes in the extent, structure,
and species composition of forests. They attributed these changes to human-induced
de-forestation and reforestation. Arneth et al. (2008, 2011) compared the responses of
the simulated BVOC emissions derived using different models to climate and
vegetation changes. They found that increasing forest area could add several tens of
percent to future isoprene emissions. Climate change could also exert influences on
isoprene emission via the changes in temperature and $CO_2$. The latter can benefit
forest productivity and leaf growth via fertilization effect. Steiner et al (2002)
simulated the effect of human induced land use changes due to urbanization and
agriculture on BVOC emissions. Their results revealed that the increasing
anthropogenic emissions of VOCs subject to urbanization overall enhanced total VOC
emissions. Most of the existing studies were carried out using climate models subject
to projected climate and land cover change scenarios.
The Three Northern Regions Shelter Forest (TNRSF) program in China, also
known as 'the Great Green Wall', began in 1978 and will terminate in 2050 (**Fig. 1**).
The program aims to increase China's forest coverage from 5% in the 1970s to 15%
by 2050. By the end of the fourth phase in 2010 of this largest afforestation program
in the human history, the vegetation coverage over the TNRSF has already reached
12.4% (Wang et al., 2011; Central Government of China, 2012). The program has
achieved great successes in mitigating local ecological environment and climate,
despite the debates on the effectiveness of the TNRSF in improving the ecological
environments in Northern China and negative influences of the program on
groundwater storage in arid and semi-arid regions (Pang, 1992; Cheng and Gu, 1992;
Parungo et al., 1994; Hu et al., 2001; Zhong et al., 2001; Ding et al., 2005; Liu et al.,
2008; Yan et al., 2011; Zheng and Zhu, 2013; Fang et al., 2001; Tan et al., 2007;
Zhang et al., 2013). Recently, the TNRSF impact on air quality was also investigated
(Zhang et al., 2015), which showed that the increased vegetation coverage in the
TNRSF has increased its efficiency in removing air contaminants from the
atmosphere as supported by the increased modeled dry deposition velocities and
fluxes of sulfur dioxide ($SO_2$) and $NO_x$ in many places of the region during the past
three decades.
Given its unique status in large-scale artificial afforestation in the human history,
the TNRSF might provide significant insights into understanding of human induced
biogenic VOC emissions on a long-term scale. In the present study, a framework
combining satellite remote sensing data, a biogenic emission model, and uncertainty
analysis was first developed to estimate BVOC emissions in Northern China.
Seasonal and annual biogenic isoprene emission inventories were then developed
from 1982 to 2010. Finally, the potential influences of the development and expansion
of the TNRSF on the long-term trends of the biogenic isoprene emissions were
investigated to discern evidence of decadal or longer-term changes in BVOC
emissions from large-scale forest restorations induced by the human activities. The
newly generated historical isoprene emissions inventories over Northern China will
also be useful for assessing past, current,and future air quality and climate issues.

**2. Methodology**
**2.1. BVOC emission model**
The MEGAN2.1 (Model of Emissions of Gases and Aerosols from Nature version 2.1)
(Guenther et al., 2012) which is an updated version of MEGAN2.0 (Guenther et al.,
2006) and MEGAN2.02 (Sakulyanontvittaya et al., 2008), was used here to estimate
BVOC emissions in Northern China. This new version includes additional compounds,
emission types, and various controlling processes. For BVOC emissions, MEGAN2.1
is primarily driven by biological and meteorological factors, including vegetation type
with which the emission factors of BVOCs are assigned, air and leaf temperatures,
light, leaf age and leaf area index (LAI), solar radiation/photosynthetically active
radiation (PAR), wind speed, humidity, and soil moisture (Guenther et al., 2006; 2012;
Pfister et al., 2008; Arneth et al., 2011). MEGAN2.1 was set up over Northern China
with a grid spacing of $0.25^{o} \times 0.25^{o}$ latitude/longitude to produce gridded daily and
monthly emission fluxes. Meteorological data used in the MEGAN2.1 employed the
6-hourly objectively analyzed data from the 1°×1° latitude/longitude NCEP (National
Centers for Environmental Prediction) Final Operational Global Analysis
(http://dss.ucar.edu/datasets/ds083.2/). These data were then interpolated into the
TNRSF grids on the spatial resolution of 0.25×0.25 latitude/longitude.PAR was
calculated from solar radiation provided by the Big-leaf dry deposition model (Zhang
et al., 2002). Twenty-two land types were used, including an additional crop type
which was not specified in the MEGAN2.1. These land types at each model grid were
identified using the surface roughness lengths estimated from satellite remote sensing
data (Zhang et al., 2015). Guenther et al. (2012) reported the differences in
MEGAN2.1 modeled annual isoprene emissions as a result of changing plant
functional type (PFT) (24 %), LAI (29 %), and meteorology (15 %) input data. This
suggests that LAI is one of crucial variables in the model.
To evaluate the MEGAN2.1 estimated isoprene biogenic emission fluxes, a field
campaign was conducted to measure total VOC (TVOC) concentrations at several
sites within and outside the TNRSF (Section 2.4). The monitored TVOC
concentrations were then converted to TVOC emission fluxes using a box model,
developed by Guenther et al (1996) which links biogenic VOC emission and
photochemical reaction with OH radicals and ozone. The model was derived from a
simplified mixed-layer scalar conservation equation, given by
$E = z_i L c$ ,                                         (1)
where $E$ and $c$ are the emission and concentration in the mixed-layer, $z_i$ is the height
of mixed-layer capping inversion, taken as 1000 m following Guenther et al (1996). $L$
is the oxidation rate of VOC subject to OH radical and ozone, defined as $[k_{OH}, OH] +$
$[k_{O3}, O_3]$, where $k_{OH}$ and $k_{O3}$ are reaction rate constants for OH and $O_3$, respectively.
The rate constants and mean concentrations of OH and ozone are presented in Table
S1 of Supplementary Materials. Further details are presented in Sections 2.4 and 3.4.
**2.2. LAI**.
LAI data with $0.25^{o} \times 0.25^{o}$ latitude/longitude resolution from 1982 to 2010 were
derived from the satellite remote sensing data of the normalized difference vegetation
index (NDVI) for the same period. Detailed descriptions of the procedures generating
LAI data for the TNRSF region were presented in Zhang et al (2015).
**2.3. Uncertainty analysis.**
Although the BVOC emissions model was well established for different vegetation
types, there were uncertainties in the estimate of BVOC emission fluxes. Some of
these uncertainties are generated from inaccurate emission factors, empirical
algorithms, and input data used in the model (Hanna et al., 2005; Guenther et al.,
2012). Situ et al showed that, in addition to the emission factors, PAR and
temperature also created large uncertainties in the MEGAN model (Situ, et al., 2014).
A Monte Carlo technique was used to evaluate uncertainties of modeled isoprene
emissions by MEGAN2.1 (Hanna et al., 2005; Guenther et al., 2006, 2012; Situ et al.,
2014). In the uncertainty analysis, each input parameter in MEGAN2.1 for isoprene
emissions, including LAI, leaf temperature (a function of air temperature), PAR,
emission factors, several empirical coefficients related to past leaf temperatures, and
solar zenith, was treated as a random variable with a normal distribution. The
MEGAN2.1 model for BVOC emissions was run repeatedly 100,000 times at the 95%
confidence level based on the coefficients of variation ($CV$, %) of these input
parameters. The Monte Carlo simulations showed that the isoprene emissions reached
approximately a normal distribution, ranging from 0.05 to 5.29 micro-mole m$^{-2}$ h$^{-1}$
with the variation from 97%-211%. Details for the uncertainty analysis are presented
in Supplementary Materials (Table S2, **Fig. S1**).
The uncertainty analysis using the Monte Carlo technique was also conducted for
the box model (Eq. 1). Analogous to the uncertainty analysis for the MEGAN2.1, this
box model was also run repeatedly 100,000 times at the 95% confidence level based
on the coefficients of variation ($CV$, %) for $z_i$, the measured isoprene concentration
($C$), and the concentrations of OH and $O_3$. The $CV$ for these four parameters were
taken from Guenther et al (1996) (Table S3). The results from Monte Carlo
simulations showed that the converted isoprene emissions from the measured
concentrations using Eq. 1 reached approximately a normal distribution, ranging from
1.2 to 152.9 μg m$^{-2}$ h$^{-1}$ with the variation from 98.3%-116.7% (**Fig. S2**).
**2.4. Ambient VOCs concentrations within and outside the TNRSF**.
As part of efforts to understand potential uncertainties in the estimation of isoprene
emissions from the TNRSF, a field campaign was conducted to measure gas-phase air
pollutants, particular matter, and persistent organic pollutants in air, foliage, and soil
within and outside the TNRSF in the summer of 2015. The first phase of this field
study focused on the Central-North China region of the TNRSF because this region
has been paid the highest attention by the TNRSF program due to its proximity to
Beijing and Tianjin, the two megacities in Northern China. Eight monitoring sites in
this region were selected, with four of these inside and another four outside the forest
(**Fig. S3**). All these sites are situated in the northwest and northeast of Beijing where
the TNRSF program was operated most successfully. Total VOC (TVOC) was
measured simultaneously using the GreyWolf TG-502/TG-503 sensors (GreyWolf
Sensing Solutions) at each paired sites within and outside the forest but on different
days at the selected 4 paired sites. The sampling frequency was set at 1 min. The
GreyWolf TG-502/TG-503 instrument uses SEN-B-VOC-PPB PID (photoionization
detector) sensor (10.6eV lamp, range: 5 to 20,000 ppb) which responds to the vast
majority of VOCs with the response time < 1 min. The environmental conditions for
sensor operating range from 0 to 90% RH (relative humidity) and from $-15^{o}$ to $60^{o}$ C.
The GreyWolf TVOC sensor adopts two points calibration approach with low point of
0 ppb and high point at 7500~9000 ppb, respectively. Standard calibration gas is
isobutylene. More details of the GreyWolf TG-502/TG-503 TVOC sensor can be
found at the GreyWolf website (https://www.wolfsense.com/directsense-tvoc-volatile-
organic-compound-meter.html). It should be noted that the GreyWolf VOC sensor can
only measure TVOC, hence the concentration of individual VOC species is not
reported here. Typical tree species planted in this region were selected in the field
monitoring program. Among them, poplars (*Populus spp*), a broadleaf tree species,
dominated the two forest sites in Langfang and northern Zhangbei County. Poplars
has been the major tree species planted across the Central-North China region of the
TNRSF over the last thirty years. From the late half of the 2000s, due to the death of
many poplars in this region, Scots pine (*Pinus sylvestris*), which is a coniferous tree
species, has been recommended and planted in this region. Scots pine is the major tree
species at northern Zhangbei County and Xinglong forest sites. As for the
comparative monitoring sites outside the forests, the Langfang site is 500 m away
from the forest and located in a corn field, the Zhangbei north and south sites are
about 1 km and 600 m, respectively, away from the forest and both are located in a
grassland, and the Xinglong site is about 400 m away from the forest and located in a
corn field. The sampling was operated in early morning from 6:15 – 8:15am, and
early afternoon from 2:15 – 4:15 pm with sampling frequency of 1 min. The sampling
date was on August 9[th], 2015 at the Langfang sites, 10[th] at the Xinglong sites, 12[th] at
the Zhangbei north sites, and 13[th] at the Zhangbei south sites. It should be noted that
this field measurement program was not aimed to determine the spatial and temporal
distributions of isoprene emissions, but instead to examine and verify the release of
this reactive biogenic VOC species from the TNRSF.
**3. Results**
**3.1. Isoprene emission inventory in TNRSF**
**Figure 2** shows the TNRSF domain-averaged annual biogenic isoprene emissions
(micro-moles $m^{-2}$ $h^{-1}$) aggregated from monthly values. The magnitudes of isoprene
emissions estimated in the present study agree with the China's BVOC emission
inventory established previously, particularly in the natural forests (Song et al., 2012;
Li et al., 2013), as  elaborated below. A long-term increasing trend up to 2007,
although with fluctuations in certain years, was observed (**Fig . 2)** The Central-North
region of the TNRSF exhibited the strongest increasing trend with the highest
emissions increase by 58% over the 30 years period.
**Figure S4** illustrates the MEGAN2.1 simulated isoprene emission fluxes across
the TNRSF in 1982, the early stage of the TNRSF construction, and 2010, the end of
the fourth phase (2001-2010) of the program, respectively. Compared with the
emission fluxes in 1982, higher isoprene emissions in the Central-North China region
and lower emission fluxes in the Northeast region and Eastern Inner Mongolia region
of the TNRSF were identified in 2010. The differences in the biogenic isoprene
emissions between 1982 and 2010 were calculated as $E_{dif} = E_{2010} - E_{1982}$. The spatial
pattern of $E_{dif}$ (**Fig. 3**) is consistent with the emission fluxes in 1982 and 2010, as
shown in **Fig. S4a** and **b**. Positive differences of $E_{dif}$ were observed in the
mountainous areas of west Xinjiang, Shaanxi, eastern Gansu provinces, and the
Central-North China region, suggesting increasing isoprene emissions associated with
the expansion of the TNRSF in these regions.
**3.2. Isoprene emission trend in the TNRSF and Northern China**
Decadal or longer time trends in isoprene emissions over the TNRSF and Northern
China can provide some insights into the impact of the large-scale artificial
afforestation on BVOC emissions - the knowledge that is needed to address air quality,
climate, and ecosystem issues. **Figure 4** illustrates modeled isoprene emission fluxes
(micro-moles m$^{-2}$ hr$^{-1}$) in 2000 (**Fig. 4a**), after 20 years construction of the TNRSF,
and the slopes (trends) of the linear regression relationship between isoprene emission
and the time sequence of 1982 through 2010 (**Fig. 4b**) over Northern China,
respectively. High isoprene emissions can be found in the regions extending from
northeast Qinghai province to Ta-Pa Mountains, the boreal forest in Northeast China,
Central-North China, and Tianshan Mountain and Pamirs in Xinjiang province. The
spatial pattern of the estimated emissions in Northeastern China is similar to Song et
al.'s results from 2008 to 2010 (Song et al., 2012). They showed high isoprene
emissions from the boreal forest in Northeastern China and Qinling – Ta Pa
Mountains.

The total annual isoprene emission, summed from annual emissions of the model

grids that fall within the TNRSF domain, ranged from  45,000 to 70,000 ton $yr^{-1}$
during 1982-2010 for the whole TNRSF (the area encircled by the blue solid line in
**Fig. 4**), and from 132,000 to 176,000 ton $yr^{-1}$ for whole Northern China (**Fig. 4**). This
is equivalent to a total emission of 1.6 Tg and 4.4 Tg, respectively, for the two regions
during the past three decades from1982 to 2010. It is worth noting that, although the
TNRSF accounts for 59% of the total area of Northern China and 42% of mainland
China (Zhang, et al., 2015), it covers almost all arid and semi-arid regions in Northern
China. Vegetation coverage in these regions was still sparse after 30 years
construction of the TNRSF, and shrubs, instead of trees, are major plant types in the
Western China region of the TNRSF. The isoprene emissions are considerably low in
these regions, as shown by **Figs. 4** and **5**. In addition, as shown by **Fig. 4**, the region
of Northern China defined in this study extends virtually to 30$^{o}$N. Although the
isoprene emissions in the TNRSF only accounted for 37% of the total emissions in
Northern China, the relatively strong increasing trend (**Fig. 2**) in the TNRSF
(slope=0.881, $R^2$=0.335) has reversed the negative trend (slope=-0.533, $R^2$=0.05) of
the total annual isoprene emissions in Northern China, which did not take the isoprene
emissions in the TNRSF into consideration, to the positive trend (slope=0.347,
$R^2$=0.014) from 1982 to 2010 in Northern China, as shown in **Fig. S5**.

To highlight the contribution of the TNRSF to the increasing isoprene emissions,

the trend of the gridded isoprene emissions over the TNRSF was further investigated.
As expected, the estimated monthly emission fluxes showed dramatic seasonal
variations with the largest values in summer and the lowest values in winter,
consistent with the seasonal changes in LAI over the TNRSF (figure not shown).
**Figure 5** presents the gridded trends of the summer biogenic isoprene emissions (Eq.
1) across the TNRSF from 1982 to 2010. The summer emission fluxes exhibit similar
annual pattern to the annual emissions (**Fig. 4b**) but are greater than the annual
emissions, as shown by **Fig. 5**. Positive trends of the emissions were observed in the
mountainous and surrounding areas of the Junggar Basins (north Xinjiang), eastern
Qinghai province in the Northwest China region of the TNRSF, the Central-North
China region, and western Liaoning province in the Northeast China region of the
TNRSF. These provinces and locations are marked in **Fig. 1**. In particular, the largest
positive trends can be observed in the areas north of the two megacities - Beijing and
Tianjin. These two megacities have been targeted as key cities to be protected by the
TNRSF from sandstorms from the north. Extensive tree planting activities have been
promoted to the north of these two megacities (Central Government of China, 2012).

**Figure 6** shows the isoprene emissions from 1982 to 2010 averaged over the

Northwest China, the Central-North China, and the Northeast China regions of the
TNRSF, respectively. It can be identified again that the domain averaged isoprene
emissions in the Central-North China region of the TNRSF exhibited a clear
increasing trend with the slope of 0.0004 ($R^2 = 0.35$, p=0.002). Whereas, statistically
insignificant and relatively weak trends of isoprene emissions were found in the
Northeastern China (slope=0.00003, $R^2$=0.032, p=0.484) and Northwestern China
(slope=0.00009, $R^2$=0.27, p=0.012) regions of the TNRSF, respectively. The increase
of isoprene emissions over the Central-North China region can be attributed to
continuous expansion of forest coverage. Compared with the Central-North region of
the TNRSF, the forests in the Northeast China region are mixed with natural forests.
These natural forests already reached the steady state before the 1980s, so they would
not contribute to the increasing trend of biogenic isoprene emissions. As shown by
**Fig. 4b**, the isoprene emissions in most places of Northeast China show almost no
trends in most places of Northeast China. The Northwest China region of the TNRSF
is arid and semi-arid area with low precipitation. Shrubs, instead of trees, were
planted in many places of this part of the TNRSF regions, resulting in low biogenic
isoprene emissions.

Trends of isoprene emissions were also compared between those within and

outside the TNRSF and in natural forests. Three small areas were selected for the
comparison, each consisting of 4 grid points, in the Central-North China region of the
TNRSF (marked by the red circle in the inner map of **Fig. 1**), a farmland outside the
TNRSF (blue circle), and in the boreal forest of Northeast China (the Greater Khingan
Mountains, marked by yellow circle in **Fig. 1**), respectively. Trends in annually
averaged isoprene emissions from these three small areas are shown in **Fig. 7.**
Significant increasing trend is only seen in the area within the TNRSF. The levels of
isoprene emissions in the other two small areas were almost uniformly distributed for
the last three decades.
**3.3. Comparison with the previous emission data**
No extensive and direct measurements of BVOC emission across the TNRSF have
been ever carried out. Several field campaigns were conducted to measure BVOC
emissions in Northern China but these monitoring programs were not typically
designated for the TNRSF (Klinger et al., 2002; Wang et al., 2003). Li et al. (2013)
established an emission inventory of BVOCs (isoprene, monoterpenes, sequiterpene
and other VOCs) over China using MEGAN2.1 model. Their results showed that
annually averaged isoprene emission fluxes ranged from  0 to 22 $\mu g\ m^{-2}\ h^{-1}$ in 2003 in
northern Xinjiang, Qinghai, Gansu, and Shaanxi provinces in the Northwest China
region of the TNRSF, and western Inner Mongolia. The average isoprene emission
fluxes estimated in the present study for the same regions and the same year ranged
from 0.01 to 18.2 $\mu g\ m^{-2}\ h^{-1}$, agreeing reasonably well with Li et al's data. Li et al's
inventory (2013) also showed high isoprene emission flux in the Central-North China
region, including the north of Shanxi and Hebei provinces, Beijing, and the natural
(boreal) forest area in Northeast China, ranging from 22 to 880 $\mu g\ m^{-2}\ h^{-1}$. While the
lower limit of their estimated flux agrees well  with our lowest emission flux of 20.4
$\mu g\ m^{-2}\ h^{-1}$, the upper limit of their emission flux was 880 $\mu g\ m^{-2}\ h^{-1}$, a factor of 4
higher than our value (122.4 μg m$^{-2}$ h$^{-1}$) for the same region. Li et al (2013) adopted
more locally updated species-specific emission factors and a vegetation classification
based on a new vegetation investigation in the late 1990s and early 2000s in China.
Their calculation also used hourly and diurnal meteorological (temperature, radiation,
winds) data. Our estimated fluxes used the emission factors specified in the
MEGAN2.1 (Guenther et al., 2012) and vegetation types classified by the roughness
lengths (Zhang et al., 2002, 2015). In addition, our model input daily meteorological
data. These different input data to the MEGAN model resulted likely in the difference
of the isoprene emission fluxes between Li et al (2013) and our results. Song et al.
(2012) simulated BVOC emissions in Eastern China from 2008 to 2010. A portion of
their model domain in Eastern China superimposed with the Central-North China and
the Northeast China region of the TNRSF defined in our study. The annually averaged
isoprene emission fluxes from 2008 to 2010 from Song et al's model simulations
ranged from 10 to 100 μg m$^{-2}$ h$^{-1}$ in Inner Mongolia region, and 100-1000 g m$^{-2}$ h$^{-1}$ in
the north of Shanxi and Hebei provinces, Beijing, and Tianjin, which were higher than
our results of 0 to 32.6 μg m$^{-2}$ h$^{-1}$ and 20.4 to 122.4 μg m$^{-2}$ h$^{-1}$, respectively, in these
two regions. Song et al. used MEGAN2.04 model with different emission factors
adjusted based on China's principal vegetation species (Song et al., 2012). These
could also lead to different biogenic isoprene emissions.
**3.4. Emissions converted from ambient concentrations**
**Figure 8** illustrates measured afternoon (local time 2-4 pm) TVOC levels in the
atmosphere at the 4 paired monitoring sites in the Central-North China region of the
TNRSF with sampling frequency of 1 min. Detailed descriptions of these sites and
sampling procedures are presented in Methodology section, **Fig. S3**, and Table S4,
respectively. Higher TVOC air concentrations were observed at all forest sites than
those sites outside the forests. In particular, the TVOC levels at the southern and
northern Zhangbei sites within the TNRSF were 3 to 4 times higher than that
measured in the grassland sites outside the TNRSF, suggesting that the forests made
significant contributions to the sampled TVOC levels. Using the box model (Eq. 1),
emission fluxes were converted from the measured TVOC concentrations at the four
forest sites. Taking the TVOC levels as the box model input (Eq. 1), and assuming
the isoprene emission to be 50% of the TVOC (Song et al., 2012; Li et al., 2013), we
obtained the emission fluxes of 32.3, 44.1, 52.9, and 44.1 $\mu g\ m^{-2}\ h^{-1}$ at the Langfang,
Xinglong, Zhangbei (North), and Zhangbei (South) sites, respectively. These values
agree nicely with the MEGAN2.1 modeled emission fluxes of 36, 41.5, 49, and 47.6
$\mu g\ m^{-2}\ h^{-1}$ at the same sites. It is noticed that the box model (Eq. 1) does not take into
account the effect of wind speed on the emissions. An effort was also made to use a
simplified Gaussian model (Eq. S1) for an area source (Arya, 1999) to convert the
measured TVOC concentrations to emissions. Under approximately calm wind
conditions ($<0.5\ m\ s^{-1}$) at the sampling sites and the same assumption of isoprene
emission as the half of the TVOC emission, the converted fluxes using this model are
about a factor of 2 higher than the MEGAN2.1 estimated fluxes. Results are presented
in Supplementary Materials. The potential differences between the MEGAN2.1
modeled and converted fluxes from the Gaussian model (Eq. S1 of Supplementary)
might be attributed to several causes. Firstly,  the TVOC concentrations were
measured at a single site within the selected forests in this field campaign which
represent typical tree species in the Central-North China region of the TNRSF.
Whereas, the underlying surface of a model grid square ($27.83 \times 27.83$ km$^2$) is not
fully covered by trees but consists of other surface types, such as croplands, bare soils,
water surfaces, and towns where BVOC emissions might be lower. In addition, in the
simplified Gaussian model (Eq. S1, Supplementary) we choose the fetch $\varDelta l$ =3km
which  is related directly to the magnitude of the converted emission fluxes which was
subject to uncertainties. Nevertheless, overall the converted fluxes from the measured
TVOC concentrations using the simplified Gaussian model are about the 2 fold of the
modeled fluxes, suggesting the reasonable accuracy of the MEGAN model applied in
the present investigation.

It is worthwhile to note that anthropogenic VOC might contribute to the ambient

concentrations of  TVOCs measured at these selected sampling sites. In addition, the
emissions and concentration ratios are not identical for all VOCs due to their
different reactivity. A VOC can be emitted in relatively low amounts but make a large
contribution to the TVOC if it is considerably less reactive than isoprene. Wang et al
(2014) collected ambient concentrations of VOCs at 27 sites across Beijing from July
2009 to January 2012,  including urban, suburban, and rural sites. To identify
potential sources of isoprene, they estimated  the ratio of isoprene to 1,3-butadiene.
While the reactivity for these two VOC compounds was similar, their emission
sources differ significantly. Vehicular exhaust was found to be the dominant source of
1,3-butadiene in Beijing (Wang et al, 2010) whereas isoprene was largely related to
biogenic emissions. Their results showed that the wintertime isoprene/1,3-butadiene
was 0.30–0.34 ppbv ppbv$^{-1}$, characterizing the emission from vehicular exhaust in
Beijing (Wang et al. 2010), suggesting that the atmospheric isoprene during the
wintertime was emitted mostly from vehicular exhaust In the warm period (May -
September), their measured ratios of isoprene/1,3-butadiene ranged from 16 to 43
ppbv ppbv$^{-1}$, two order of magnitude higher than that in the wintertime, indicating that
the summertime isoprene was released from biogenic sources. Considering that our
sampling sites (especially the Langfang and Xinglong sites) are close to Beijing and
covered by similar tree species to those planted in the suburban and rural areas of
Beijing, the results from Wang et al (2014) might be applicable in our cases because
our measurements were also taken in the summertime (August). In particular, our
sampling sites are all located in rural areas, far away from traffic, industrial, and
residential areas, indicating weak influence of the anthropogenic emissions on the
measured TVOC level, half of which has been hypothesized to be isoprene in the
present study.
**4. Discussions**
Overall the estimated biogenic isoprene emission fluxes across the TNRSF illustrated
an increasing trend from the 1980s onward (**Fig. 2**). The incline trend was most
significant in the Central-North region of the TNRSF where most intensive
afforestation has been carried out in Northern China (Zhang and Zhu, 2013), in order
to protect the national capital (Beijing) region from dust and sandstorms. The
increasing biogenic isoprene emissions can be attributed to the development of the
TNRSF. The forest expansion in the TNRSF can be identified by the satellite derived
LAI, as seen from **Fig. S6a** and **b**. The linear increasing trend of the LAI across the
TNRSF is consistent with the modeled isoprene emission fluxes. The maximum
increase (58%) of the isoprene emissions from 1982 to 2010 in the Central-North
region of the TNRSF seems to agree well with the model prediction by Arneth et al.
(2008, 2011) based on projected land use changes. Their modeling results suggested
that increasing forest area could lead to several tens of percent change in biogenic
isoprene emissions.

As shown above, the significant incline trend of the annual total isoprene

emissions in the TNRSF has affected the long-term trend of the emission in Northern
China.  This implies that the increasing emission trend across the TNRSF could alter
the large-scale BVOC emissions not only in the TNRSF, but also in Northern China.
Considering that the TNRSF occupies 59% of Northern China and 42% of whole
mainland China. Future impacts of the TNRSF on BVOC emissions may be even
stronger with continuous  increase of vegetation coverage till the end of the program
in 2050.

While BVOC emissions vary on short time scales, the global BVOC emissions

are often assumed to change little on long-term (e.g., decadal) scale (Purves et al.,
2004; Sindelarova et al., 2014) considering the steady state of global forests. Since
BVOCs can partition onto or form particles in the atmosphere after oxidation, their
emissions could affect aerosol formation, cloud condensation nuclei, and climate
(Makkonen et al., 2012, Penuelas and Staudt, 2010). Identification of the impact of
climate change on BVOC emissions is not straightforward if regional or global
forests reach a steady state. The evidence identified in this study suggested that the
human-induced BVOC emissions via large-scale afforestation exert strong influence
on long-term BVOC emission and should be taken into consideration in projected
climate change scenarios, at least on a regional scale, such as Northern China. As a
precursor of secondary organic aerosols and tropospheric ozone, the significant
incline of biogenic isoprene emissions also carry significant implications to the air
quality in Northern China. Heavy air pollutions in Beijing-Tianjin-Hebei (**Fig. 1**) have
been widely known nationally and internationally, characterized by year round high
levels of fine particular matter ($PM_{2.5}$) and high surface ozone concentrations in the
summertime. Chinese government has decided to extend the TNRSF as one of the
primary measures to reduce and remove air pollutants from Beijing-Tianjin-Hebei
area (Chinese Environmental Protection Agency, 2013). As shown in **Figs. 5** and **6**,
the TNRSF in the Central-North region covering a large part of Beijing-Tianjin-Hebei
area has already gained the most rapid development as compared to the other two
northern regions of the TNRSF (**Fig. 1**), leading to marked incline of isoprene
emissions. However, it is not yet clear if and how the extension of the TNRSF could
otherwise improve local air quality. Our previous study suggested that the TNRSF
played a moderate role in removing $SO_2$ and $NO_x$ (Zhang et al., 2015). Under the
rapidly increasing $NO_x$ emissions in the past decade due to rapidly increasing number
of private vehicles in Beijing-Tianjin-Hebei area, it is necessary to assess the
interactions between BVOC emissions from the TNRSF and local air quality in this
region.

In addition to its long-term trend, isoprene emission also exhibited short-term

interannual fluctuations, as also observed from **Fig. 2**. Factors causing the fluctuations
or interannual changes in the emission fluxes depend on meteorological and
biological processes. Afforestation and deforestation often took place during the
course of the TNRSF construction due to favorable or unfavorable weather and
climate conditions for tree growth. For example, 10% - 50% of trees planted since the
late 1970s in the Central-North region of the TNRSF were reported dead since 2007
(Zhang et al., 2013; Tan and Li, 2015), causing visible decline of the forest coverage
and isoprene emissions in this region after 2007, as shown in **Fig. 2**. The lower
isoprene emission in 2010 in the Northeast China region and eastern Inner Mongolia
region of the TNRSF as compared with that in 1982 was inconsistent with the
increasing trend of the emission. The forest coverage in the Northeast China region
did not show considerable change between 1982 and 2010. On the other hand, lower
annual temperatures (e.g., by around $1^{\circ}C$) in 2010 than in 1982 were evident over the
Northeast China region of the TNRSF (**Fig. S7a**), which likely caused lower biogenic
emissions in 2010 (Purvis et al., 2004; Arneth et al., 2008, 2011). In addition,
compared with the increasing trend of LAI in the Northeastern China region of the
TNRSF (**Fig. S6a**), no statistically significant increasing trends of the isoprene
emissions are discerned in this region. **Figure S7b** displays the trend of annual
surface air temperatures (SAT, $^{\circ}C$) in the Northeast China region of the TNRSF from
1982 to 2010. Overall the SATs exhibited a decreasing trend, caused mostly by
declining SATs since the late 1990s. Since temperature plays a key role in canopy
BVOC emissions (Guenther et al., 2012; Li et al., 2013), the lack of the incline trend
of the isoprene emission fluxes in the Northeastern China region of the TNRSF might
be attributable to the decreasing SAT from the late 1990s. Another environmental
factor that may exert the influence on the trend of isoprene emissions is solar
radiation/PAR (Situ et al., 2014). Analogous to the response of the BVOC emissions
to temperature, increasing radiation could also enhance the isoprene emissions, or
vice versa, particularly on daily or monthly basis.

The comparison between the isoprene emission trends and the emissions in

2000 in Northern China also carries a significant implication for the human induced
BVOC emissions. As shown from **Fig. 4b**, the trend of isoprene emissions from 1982
to 2010 over Northern China showed a rather different spatial pattern from its
emissions in 2000 (**Fig. 4a**). No significant trends were observed in the boreal forest
in Northeastern China,though a larger amount of isoprene was emitted from the
forest in this region in 2000. This implies that this natural forest was likely under a
steady state from which the biogenic isoprene emissions were not altered on the
decadal basis (Sanderson et al., 2003; Purves et al., 2004).

Although Qinghai – Ta-Pa Mountains exhibited the highest emissions in 2000

(**Fig. 4a**), negative trends of the biogenic isoprene emissions dominated this area,
indicating the declining of the emissions over the period of 1982 through 2010. This
is consistent with the decreasing vegetation coverage during this period in this region,
as shown by the negative trends of the leaf area index (LAI) in Northern China (**Fig.**
**S6**). On the other hand, most positive trends can be identified in the Central-North
region and along the foots of Tianshan Mountain in west China (see the areas
encircled by the solid blue line in **Fig. 4)**. This manifests that the TNRSF exerts strong
influences on biogenic VOC emissions, particularly on their decadal variation, though
the magnitude of emissions might not be higher than that  from natural forests in
Northeastern China (**Fig. 4a**). Results further imply that the TNRSF is very likely the
major source contributing to the increasing biogenic isoprene emissions over the past
30 years and many years to come in Northern China. Climate change has been
thought also to play an important role in the changes in biogenic emission of isoprene
on decadal or longer time scale because it can alter temperature and vegetation
coverage (Turner et al., 1991; Sanderson et al., 2003). It is unknown if and to what
extent the increasing vegetation coverage and temperature over the TNRSF were
induced by climate change. Evidence shows that the human induced afforestation
contributed mostly to the increased vegetation coverage over the TNRSF and
Northern China (Wang et al., 2011), as shown by **Fig. S6a**, and hence to the increased
biogenic isoprene emissions
Among the three small areas within the TNRSF, in the farmland, and in the
boreal forest of Northeastern China (**Fig. 7**), the emission values increased by nearly 5
times from 1982 to 2010 in the area within the TNRSF with the slope of 0.0018 ($R^2$ =
0.55). On the other hand, no statistically significant increasing trends of biogenic
isoprene emissions were found in the farmland and the boreal forest, though the
higher emissions were observed in the boreal forest. More interestingly, the biogenic
isoprene emissions in the selected small area of the Central-North China region tend
to surpass the isoprene emissions in the boreal forest from 2004 onward. This can be
partly attributed to rapidly growing forest coverage and higher temperatures in this
region as compared to Northeastern China. The large area of foliage trees planted in
this region also played a role for relatively high and increasing isoprene emissions as
compared with the boreal forests in Northeast China where coniferous trees are major
tree species which release relatively lower isoprene to the atmosphere as compared to
broadleaf trees in the selected area in the Central-North China region of the TNRSF
(Guenther et al, 2012).
**5**. **Conclusions**
Gridded monthly and annual biogenic isoprene emissions in Northern China were
modeled for the period of 1982 to 2010 and were then applied to assess the long-term
trends of the biogenic isoprene emissions in the TNRSF in order to discriminate the
signals of the human activities in decadal and longer-term trends of BVOCs on large
spatial scales. Significant impacts of the TNRSF on the BVOC emissions in Northern
China were identified during the past three decades. Annual isoprene emissions in
many places of the TNRSF region, especially in the Central-North China region,
exhibited an inclining trend. The maximum increase in the isoprene emission flux
reached 58% between 1982 and 2010, indicating important roles of the human
activities on BVOC emissions. The comparison of isoprene emission fluxes among
the Central-North China region of the TNRSF, farmland, and the boreal forest in
Northeastern China outside the TNRSF revealed that the biogenic isoprene emissions
in some areas of the Central-North China region of the TNRSF produced by man-
made forests have surpassed the emissions from the natural forests. This suggests that
the TNRSF was a main contributor to the decadal or longer-term changes in BVOCs
in Northern China. The impact of the TNRSF on BVOC emissions is expected to be
stronger in the coming years along with continuous development of the TNRSF
program till 2050. Since VOCs are major precursor of tropospheric ozone, future
studies are needed to investigate how the increased BVOCs in the TNRSF contribute
to ozone formation, especially in the case of concurrently increasing $NO_x$ emissions in
Northern China.
**The Supplement related to this article is available online.**
**Acknowledgement**
This work is supported by the National Natural Science Foundation of China through
grants 41371478 and 41371453.

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

**Figures captions**
**Figure 1.** The Three Northern Regions Shelter Forest (TNRSF) in Northern China.
The Northwest China region of the TNRSF, defined by grey color, includes Xinjiang,
Gansu, the north of Qinghai, Ningxia, West Inner Mongolia, and the north of Shaanxi;
The Central-north China region, defined by orange gold color, includes the north of
Shanxi and Hebei provinces, Beijing, Tianjin, and Central Inner Mongolia; The
Northeast China region, defined by brass color, includes East Inner Mongolia, part of
Liaoning, Jilin, and Heilongjiang provinces. Red, blue and yellow circles in the inner
figure (right-lower corner of the figure) indicate three small areas in the TNRSF, a
farmland, and the boreal forest from which isoprene emission flux are extracted for
comparison (see Results and Discussions sections). Two megacities, Beijing and
Tianjin in the Central-North China region, are also indicated.
**Figure 2**. Domain-averaged annual emission flux (micro-moles $m^{-2}$ $h^{-1}$) of isoprene
over the TNRSF from 1982 to 2010. Red dot line indicates linear trend of emission
fluxes and shading stands for $\pm$ 1 standard deviation of emission fluxes.
**Figure 3.** Differences of emission flux ($E_{2010}$ - $E_{1982}$, micro-moles m$^{-2}$ h$^{-1}$) of isoprene
between 1982 and 2010. The emission fluxes in these two years are shown in Fig. S3a
and b of Supporting Information
**Figure 4.** (a) Gridded annual isoprene biogenic emission (micro-moles m$^{-2}$ h$^{-1}$) in
2000 over Northern China with spacing 1/4$^o$ × 1/4$^o$ latitude/longitude; (b) slopes of
linear regression relationships between annual mean isoprene emission flux (micro-
moles m$^{-2}$ h$^{-1}$) and the time sequence (or linear trend) from 1982 to 2010 across
Northern China.
**Figure 5.** Slopes of linear regression relationships between summer mean isoprene
emission flux (micro-moles m$^{-2}$ h$^{-1}$) and the time sequence (or linear trend) from 1982
to 2010 across the TNRSF.
**Figure 6.** Annual variations of emission fluxes of isoprene averaged over three
regions of the Northeast, Central-North, and Northwest China region of the TNRSF.
Dotted straight line represent linear trend of isoprene emission fluxes in the Central-
North China region.
**Figure 7.** Annual variation and trend of isoprene emission flux spatially averaged
over three small areas in and outside the TNRSF in Central-North China and natural
(boreal) forest region as marked in **Fig. 1**. The left-hand-side y-axis scales trend of
isoprene emission fluxes in the TNRSF region and boreal forest in Northeast China
and right-hand-side y-axis scale emission flux from the farmland outside the TNRSF.
**Figure 8.** Measured ambient concentrations of TVOC (mg m$^{-3}$) with frequency of 1
min from 2 – 4 pm local time at 4 paired monitoring sites within and outside the
TNRSF. (a) Langfang (August 9 2015), (b) Xinglong (August 10 2015); Zhangbei
(North, August 12 2015), (c) Zhangbei (South, 13 August 2015).

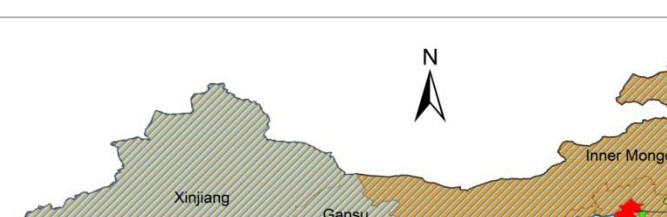
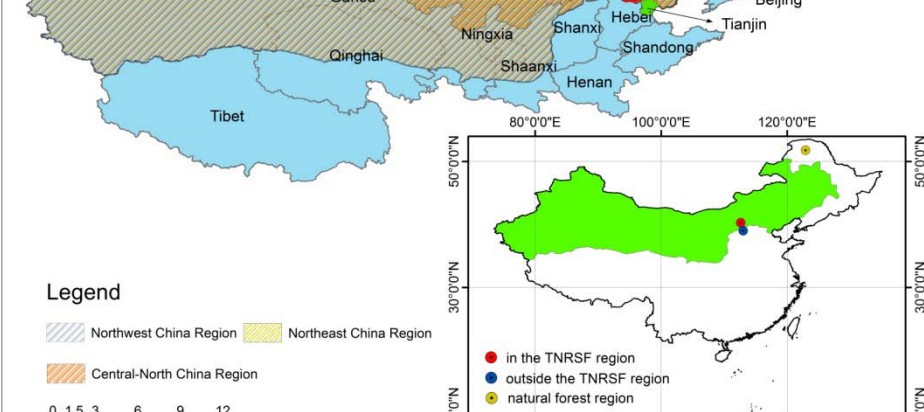

**Figure 1**

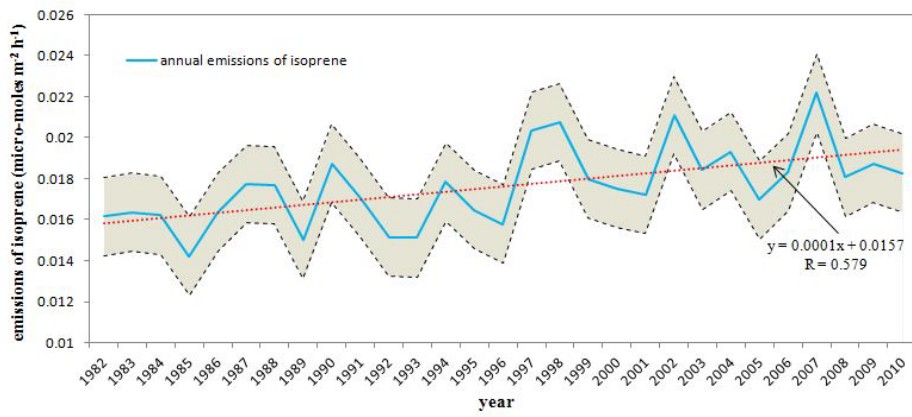

**Figure 2**

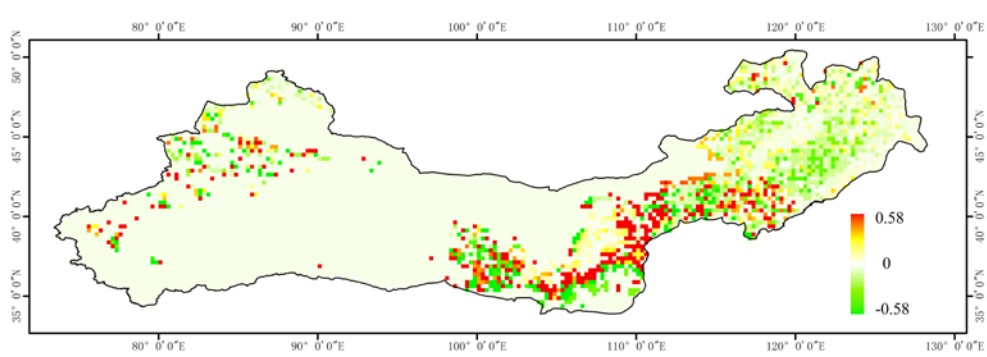

**Figure 3**

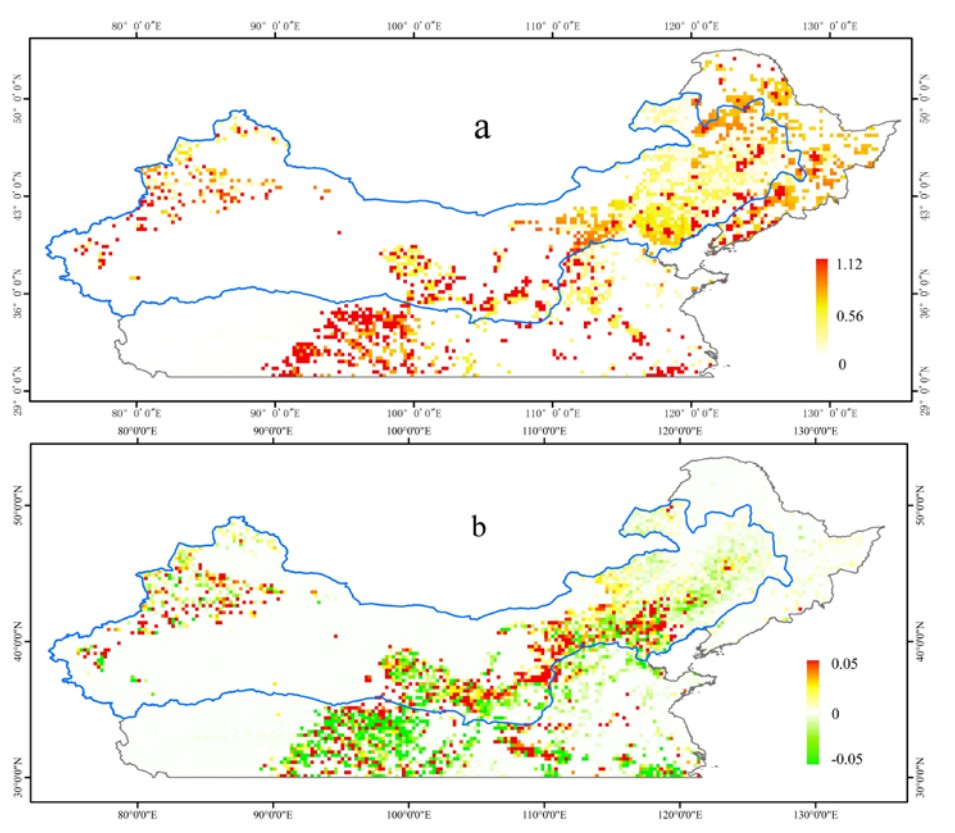

**Figure 4**

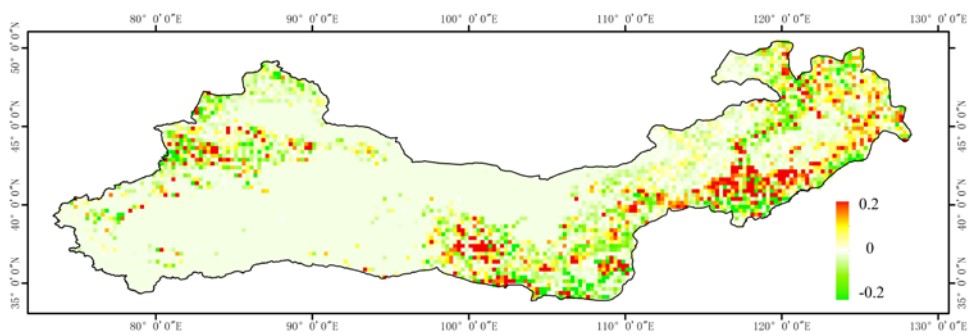


**Figure 5**

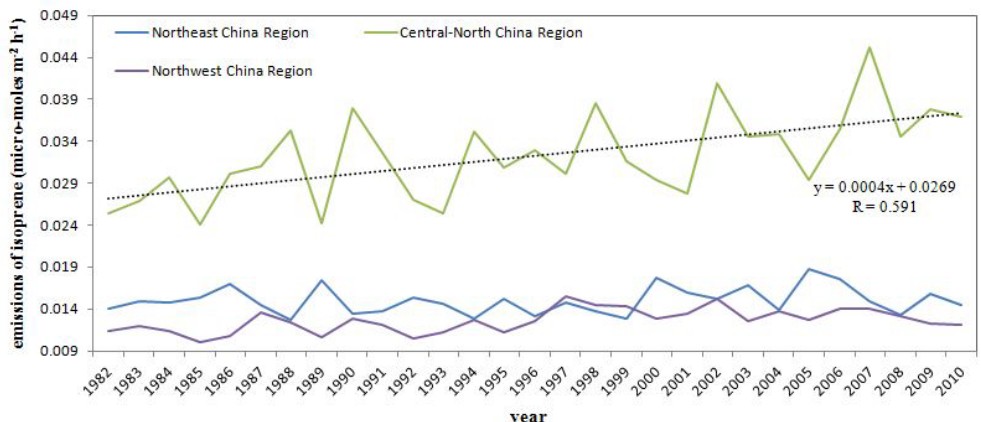


**Figure 6**

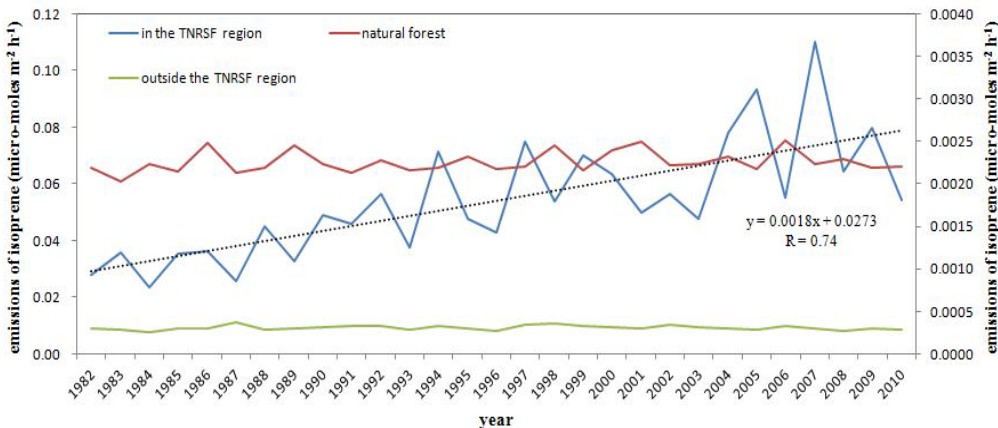


**Figure 7**

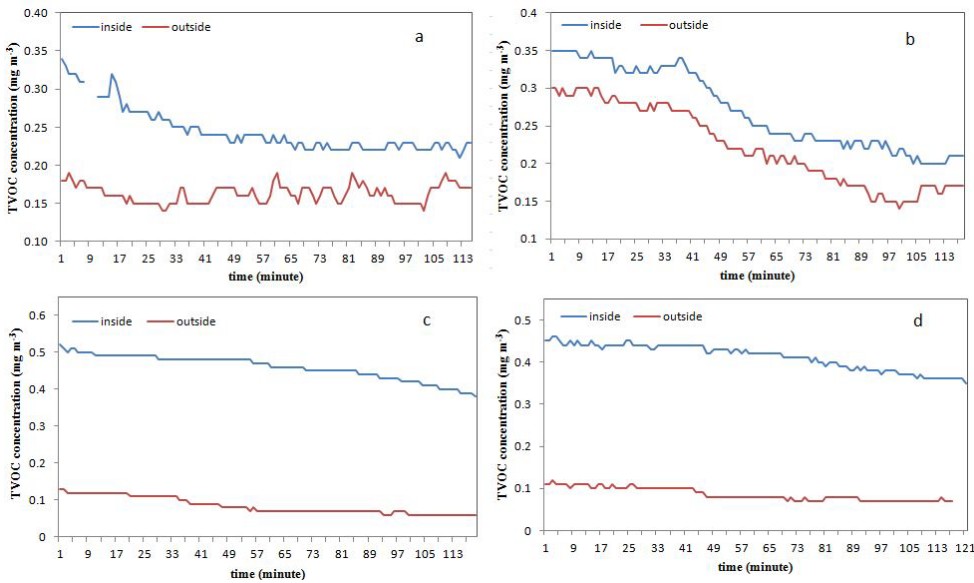

**Figure 8**