# Peer review of "increasing trend of biogenic isoprene emissions in Northern China"

_Atmospheric Chemistry and Physics, 2015_

## Referee Comment (RC1) · K. Sindelarova (Referee) · 25 Feb 2016

General comments

The paper touches an interesting topic of impact of human induced ecosystem changes on air quality. It studies trends of isoprene emission in Northern China with focus on artificially grown ecosystem of Three Northern Regions Shelter Forest (TNRSF). By using model of biogenic VOCs the paper shows that there is an increasing trend in biogenic isoprene emissions in the TNRSF region over the period of 1982 to 2010, which is likely to increase with further plantation of this human induced forest. Particularly, the study shows that promoted tree plantation in Central-North China region close to agglomerations of Beijing and Tianjin brings higher isoprene emissions to the vicinity

of potentially strong NOx sources, which can have significant impact on local air quality (e.g. surface ozone).

The base of the study is in modeling of isoprene emissions with the MEGAN model (Guenther et al., 2012). Additionally, the authors perform an uncertainty analysis of model inputs using the Monte Carlo method. And furthermore, they carry out a model evaluation by converting the measurements of total VOCs (TVOC) at several stations inside the forest to emissions of isoprene. Although the applied methodology of estimation of isoprene emission fluxes from measured TVOC concentrations is rather approximative, it provides a qualitative evaluation of modeled isoprene emissions.

The paper is comprehensibly structured, written in appropriate level of English. I recommend its publication in ACP after minor revisions. Please see my specific comments and technical corrections below.

Specific comments

1] Since the manuscript does not show any results on the impact of BVOC emissions on the 'ozone formation', it should not be included among the Key words.

2] P2L10 : "... emit harmful gases into the air ... " – gases that trees emit are not harmful per se, but they indeed contribute to air pollution through atmospheric chemistry. Please rephrase this sentence.

3] In section 2.1 authors do not mention which meteorological fields they used to drive the MEGAN model.

4] P11L7-L9 : Comparison of Northern China emissions with emissions calculated for regions in the US. This sentence is a bit vague. Authors should specify why they chose the US regions for comparison and be more precise, e.g. add numbers of total amounts.

5] It would be helpful for orientation in the Northern China geography, if the figures with maps of emissions and emission trends (Figs. 3-5, S4, S6a, S7) included indications

of meridians and parallels of latitude in the model domain.

6] It is not quite clear what is shown in Fig. 5. The description in the main text (P14L1) is not clear and the figure caption is the same as in Fig. 4b.

7] P16L4 – Authors compare their results of isoprene emissions in Central-North China regions with emission estimates by Li et al. (2013). They claim the results are comparable. However, the upper limit of their emission range is about 4 times lower than that of Li et al. (2013). Can authors comment on that, what could be the possible differences?

8] P22L2 – Discussion of the comparison of emissions in Northeast China and Inner Mongolia in 2010 and 1982 doesn't seem to be correct. While I agree with the authors' conclusion that emissions are lower in 2010 than in 1982 (shown in Fig. 3) due to lower air temperatures (Fig. S7), the premises seem to be confused. The emissions in this region do not have a trend (as shown in Fig. 4b and Fig. 6), but the forest coverage increased between 1982 and 2010 (Fig. S6a). Assumption that the mixed forest reaches a steady state is unclear though. Can you be more specific?

Technical corrections

Main text:

In the whole text please replace 'BVOCs emissions' by 'BVOC emissions', similarly 'VOCs emissions' by 'VOC emissions'

P2L17: replace 'anthropogenic emissions' by 'anthropogenic sources'

P2L18: isoprene is a subgroup of terpenes (hemiterpene), please replace 'terpenes' by 'monoterpenes' or 'monoterpenes and sesquiterpenes'

P3L10: replace 'monoterpene' by 'monoterpenes'

P3L14: replace 'was from isoprene emission' by 'was isoprene'

P4L10: replace 'tens percent' by 'tens of percent'

P5L11: replace 'modeled increased dry deposition' by 'increased modeled dry deposition'

P11L14: replace 'increased' by 'increase'

P12L1 : reference to Fig.1 seems to be redundant

P12L5: reference to Fig.3 seems to be redundant

P13L1: I'd recommend to replace 'applicable model grids' by 'model grids that fall within the TNRSF domain' or similar

P13L7: Fig.1 is not the right reference here since it does not show arid or semi-arid regions

P19L15: misspelled reference of Arneth et al.

P19L17; replace 'tens percent' by 'tens of percent'

P21L7: Sentence starting 'However, . . .' does not make sense. Did the authors mean 'However, it is not yet clear . . .' ?

P22L4: Missing space in 'between1982'

P22L14: Reference to Fig. 4b is misleading here. Either remove it, or refer to Fig. 4b directly after 'Northeastern China' in the same sentence and refer to Fig. 4a after '2000'.

References:

- missing year of publication for Guenther et al., Estimates of regional natural volatile organic compound fluxes from enclosure and ambient measurements.

Figure caption to Fig. 6 – please edit the text, only one dotted line is shown in the figure.

Supplementary material:

- in section of Simplified Gaussian model for an area source – variable 'Cis' is not defined

Figure caption to Fig. 6b – replace 'LAT' by 'LAI'

---

## Referee Comment (RC2) · Anonymous Referee #2 · 2 Mar 2016

Zhang et al have made MEGAN model simulations of isoprene emissions in China for the period 1982-2010, with special emphasis on the effects of the massive afforestation currently underway in the Three Northern Regions Shelter Forest (TNRSF) area. Model simulations showed an increase of isoprene fluxes over the years in the areas where forested cover also increased, suggesting that the man-made afforestation played a major role in the change of isoprene emissions.

This paper deals with the impact of human activities on the vegetation cover of a big land area that in turn impacts the concentrations of isoprene, an atmospherically relevant volatile organic compound that participates in the photochemistry of the atmosphere and can have an active role in the pollution episodes that China has been suffer-

ing in recent years. Thus this paper is within the scope of ACP and I would recommend its publication after addressing some concerns. The text needs some rewriting to make it clearer to the reader, especially the part reporting the TVOC measurements and the modeling of fluxes from those measurements.

Specific comments:

P2L2: correct the number "R2=0014", there must be a decimal point missing.

P2L10: defining reactive BVOCs emitted by plants as "harmful gases" is not appropriate. Authors can argue that they have implications for atmospheric generation of pollutants such as ozone, but not that these gases are harmful.

P6L19: Reference to Guenther et al 2006, is it correct? If the MEGAN version was 2.1, should this reference be Guenther et al 2012? Otherwise, MEGAN version should be 2.0.

P9L1: Should L (oxidation rate) be replaced with C (isoprene concentration) in the text? Table S3 does not list L but C, and it is reasonable that L will actually vary with OH and O3 concentrations, which are also listed in this list.

P10L7: Please write the genus name Populus starting with capital letter. Was it only one species of poplars that were planted in the region? If so, please give the scientific name, otherwise list as Populus spp and refer to this trees in plural in the text.

P10L10: Please list the variety of P. sylvestris or otherwise remove the word "var".

P13L13: The slope of -0.534 applies to northern China without including the TNRSF, according to Fig S5. Please clarify in the text.

P14L14: Did the authors do any statistical analysis to support the statement that the trends of isoprene emissions in the Central-North region are statistically significant whereas those from the other two regions are not?

P17L8-P18L2: Please clarify this part of the text. If the surface of a model grid square

is not completely covered by vegetation, wouldn't this imply that the calculated MEGAN fluxes do not compare so nicely with the estimates using Eq (1), mainly because the MEGAN fluxes calculated for these sites where TVOC measurements were performed would be higher (more vegetation coverage than the model grid square)?

P18L14: was vehicular exhaust the dominant source of atmospheric isoprene? Do the authors want to say that vehicular exhaust was the dominant source of atmospheric VOCs? Same for line 17 of this page.

P19L15: Correct Arneths to Arneth.

P20L10-13: Please clarify this sentence.

P22L1: This sentence needs more information to make sense. As it currently reads, it may seem that 2007 was a bad year for the trees, but looking at Fig. 2, isoprene emissions are at or near the historical maximum. I suspect the authors have something else in mind that is not clear to me. What is the time span that the authors describe as showing a "considerable decline of forest coverage and isoprene emissions"?

P22L7-8: the authors assume steady state of the mixed forest of Northeast China, regarding which variable? LAI? If so, have the authors checked whether the LAI information on Fig S6 agree with this assumption?

P27L10: Please list the year of publication (1996) and the complete list of authors.

FigS6 (caption): LAT should be LAI?

---

## Author Comment (AC1) · 7 Mar 2016

Responses to reviewer's comments: Dr. K. Sindelarova, Reviewer #1

General comments The paper touches an interesting topic of impact of human induced ecosystem changes on air quality. It studies trends of isoprene emission in Northern China with focus on artificially grown ecosystem of Three Northern Regions Shelter Forest (TNRSF). By using model of biogenic VOCs the paper shows that there is an increasing trend in biogenic isoprene emissions in the TNRSF region over the period of 1982 to 2010, which is likely to increase with further plantation of this human induced forest. Particularly, the study shows that promoted tree plantation in Central-North China region close to agglomerations of Beijing and Tianjin brings higher isoprene

emissions to the vicinity of potentially strong NOx sources, which can have significant impact on local air quality (e.g. surface ozone).

The base of the study is in modeling of isoprene emissions with the MEGAN model (Guenther et al., 2012). Additionally, the authors perform an uncertainty analysis of model inputs using the Monte Carlo method. And furthermore, they carry out a model evaluation by converting the measurements of total VOCs (TVOC) at several stations inside the forest to emissions of isoprene. Although the applied methodology of estimation of isoprene emission fluxes from measured TVOC concentrations is rather approximative, it provides a qualitative evaluation of modeled isoprene emissions. The paper is comprehensibly structured, written in appropriate level of English. I recommend its publication in ACP after minor revisions. Please see my specific comments and technical corrections below.

Response: We are very grateful for Dr. Sindelarova's detailed advice and constructive comments on the manuscript which benefit to the significant improvements of this paper. We agree with all of the suggested revisions and comments from the reviewer. Following the comments from Dr. , Sindelarova (Reviewer #1), we have revised the manuscript and address all comments from Dr. , Sindelarova. Our detailed responses and revisions in accordance with Dr. Sindelarova's comments are presented below. . Specific comments 1] Since the manuscript does not show any results on the impact of BVOC emissions on the 'ozone formation', it should not be included among the Key words.

Response: 'ozone formation' was removed from the key words.

2] P2L10 : "... emit harmful gases into the air ... " – gases that trees emit are not harmful per se, but they indeed contribute to air pollution through atmospheric chemistry. Please rephrase this sentence.

Response: In the revised manuscript, we have rewritten text as "they also contribute to air pollution through atmospheric chemistry".

3] In section 2.1 authors do not mention which meteorological fields they used to drive the MEGAN model.

Response: In addition to air temperature mentioned previously in our paper, we have added "solar radiation, wind speed, humidity," in the revised paper.

4] P11L7-L9 : Comparison of Northern China emissions with emissions calculated for regions in the US. This sentence is a bit vague. Authors should specify why they chose the US regions for comparison and be more precise, e.g. add numbers of total amounts.

Response: We thought that MEGAN model has been applied extensively in the US. The results from the MEGAN modeling in the US might be used as a reference to validate our modeling results in China. Nevertheless, the text on the comparison of isoprene emissions between the US and Northern China have been deleted in the revised paper.

5] It would be helpful for orientation in the Northern China geography, if the figures with maps of emissions and emission trends (Figs. 3-5, S4, S6a, S7) included indications of meridians and parallels of latitude in the model domain.

Response: Following the reviewer's suggestion, meridians and parallels of latitude have been presented in the revised Figs. 3-5, S4, S6a, and S7, respectively.

6] It is not quite clear what is shown in Fig. 5. The description in the main text (P14L1) is not clear and the figure caption is the same as in Fig. 4b.

Response: There was indeed an error in the description and caption of Fig. 5. Figure 5 shows summer gridded trends of isoprene emissions whereas Figure 4b shows the annual trends. In the revised paper we have replaced "annual biogenic isoprene emissions" by "summer biogenic isoprene emissions". The same change was made in Fig. 5 caption. We have also added new text, indicating that the summer emission fluxes 'show a similar annual pattern to the annual emissions (Fig. 4b) but are greater than

the annual emissions, as shown by Fig. 5" in the revised paper.

7] P16L4: Authors compare their results of isoprene emissions in Central-North China regions with emission estimates by Li et al. (2013). They claim the results are comparable. However, the upper limit of their emission range is about 4 times lower than that of Li et al. (2013). Can authors comment on that, what could be the possible differences?

Response: We agree with the reviewer's comment. In the revised paper we acknowledged the difference of the upper limits of isoprene emissions between Li et al (2013) and our results and listed potential reasons causing this difference. Li et al (2013) adopted more locally updated species-specific emission factors and a vegetation classification based on a new vegetation investigation in the late 1990s and early 2000s in China. Their calculation used hourly and diurnal meteorological (temperature, radiation, winds) data. Our estimated fluxes were obtained using the emission factors specified in the MEGAN2.1 (Guenther et al., 2012) and vegetation types classified by the surface roughness lengths (Zhang et al., 2015). In addition, our model input daily meteorological data. These different input data to the MEGAN model likely resulted in the difference of the isoprene emission fluxes between Li et al (2013) and our result. These texts have been incorporated into the revised manuscript.

8] P22L2 – Discussion of the comparison of emissions in Northeast China and Inner Mongolia in 2010 and 1982 doesn't seem to be correct. While I agree with the authors conclusion that emissions are lower in 2010 than in 1982 (shown in Fig. 3) due to lower air temperatures (Fig. S7), the premises seem to be confused. The emissions in this region do not have a trend (as shown in Fig. 4b and Fig. 6), but the forest coverage increased between 1982 and 2010 (Fig. S6a). Assumption that the mixed forest reaches a steady state is unclear though. Can you be more specific?

Response: Reviewer raised a good question! We agree with the reviewer that, compared with the increasing trend of LAI in the Northeastern China region of the TNRSF

(Fig. S6a), no statistically significant increasing trend of the isoprene emissions are discerned in this region. In addition to the LAIs, isoprene emissions also respond to light and temperature in terms of the MEGAN model. We further estimated the trends of gridded surface air temperatures (SATs, C) over the Northeastern China region of the TNRSF from 1982 to 2010. The result is presented in a new figure (Figure S7b, attached with this respond as Fig. 1) in the revised Supplementary. As shown, compared with the Central-North China region, the SATs in most places of the Northeastern China region exhibit a declining trend during this period of time. Since BVOC emissions are highly sensitive to changes in ambient temperatures (Guenther et al., 2012; Li et al., 2013), the lack of the incline trend of the isoprene emission fluxes in the Northeastern China region might be attributable to the decreasing SATs from 1982 to 2010. The above argument and point have been incorporated into the revised manuscript. In the revised manuscript, we have removed text on steady state of mixed forest in the Northeastern China region.

Technical corrections Main text: In the whole text please replace 'BVOCs emissions' by 'BVOC emissions', similarly 'VOCs emissions' by 'VOC emissions'

Response: Done! Thanks!

P2L17: replace 'anthropogenic emissions' by 'anthropogenic sources'

Response: Done!

P2L18: isoprene is a subgroup of terpenes (hemiterpene), please replace 'terpenes' by 'monoterpenes' or 'monoterpenes and sesquiterpenes'

Response: Following the reviewer's suggestion, 'terpenes' has been replaced by 'monoterpenes'.

P3L10: replace 'monoterpene' by 'monoterpenes'

Response: Following the reviewer's suggestion, 'monoterpene' has been replaced by 'monoterpenes' in the revised paper.

[Figure]

P3L14: replace 'was from isoprene emission' by 'was isoprene'

Response: Done!

P4L10: replace 'tens percent' by 'tens of percent'

Response: Done!

P5L11: replace 'modeled increased dry deposition' by 'increased modeled dry deposition'

Response: Done!

P11L14: replace 'increased' by 'increase'

Response: Done!

P12L1 : reference to Fig.1 seems to be redundant

Response: Reference to Fig. 1 has been deleted.

P12L5: reference to Fig.3 seems to be redundant

Response: Reference to Fig. 3 has been deleted.

P13L1: I'd recommend to replace 'applicable model grids' by 'model grids that fall within the TNRSF domain' or similar

Response: Following the reviewer's comment, 'applicable model grids' has been replaced by 'model grids that fall within the TNRSF domain' in the revised manuscript.

P13L7: Fig.1 is not the right reference here since it does not show arid or semi-arid regions.

Response: Reference to Fig. 1 has been deleted.

P19L15: misspelled reference of Arneth et al.

Response: Corrected. Thanks!

P19L17; replace 'tens percent' by 'tens of percent'

Response: Done! Thanks for the correction.

P21L7: Sentence starting 'However,...' does not make sense. Did the authors mean 'However, it is not yet clear ...' ?

Response: Reviewer is right! 'yet' is 'not yet'. We have corrected this typo error.

P22L4: Missing space in 'between1982'

Response: Thanks to the reviewer for the correction! The space between two words has been added.

P22L14: Reference to Fig. 4b is misleading here. Either remove it, or refer to Fig. 4b directly after 'Northeastern China' in the same sentence and refer to Fig. 4a after '2000'.

Response: Following the reviewer's suggestion, we have removed the reference to Fig. 4b in the revised paper.

References: - missing year of publication for Guenther et al., Estimates of regional natural volatile organic compound fluxes from enclosure and ambient measurements.

Response: Missing year and full authors list have been added in the revised reference.

Figure caption to Fig. 6 – please edit the text, only one dotted line is shown in the figure.

Response: Following the reviewer's comment we have edit the text and made corresponding changes in the figure caption of Fig. 6.

Supplementary material: - in section of Simplified Gaussian model for an area source – variable 'Cis' is not defined.

Response: Cis has been defined in the revised Supplementary material.
Figure caption to Fig. 6b – replace 'LAT' by 'LAI'

Response: Yes LAT is 'LAI'. This error has been corrected. Thanks!

[Figure]

Northeastern China Region

**Fig. 1.**

---

## Author Comment (AC2) · 7 Mar 2016

Responses to reviewer's comments Anonymous Reviewer #2

Zhang et al have made MEGAN model simulations of isoprene emissions in China for the period 1982-2010, with special emphasis on the effects of the massive afforestation currently underway in the Three Northern Regions Shelter Forest (TNRSF) area. Model simulations showed an increase of isoprene fluxes over the years in the areas where forested cover also increased, suggesting that the man-made afforestation played a major role in the change of isoprene emissions.

This paper deals with the impact of human activities on the vegetation cover of a big

land area that in turn impacts the concentrations of isoprene, an atmospherically relevant volatile organic compound that participates in the photochemistry of the atmosphere and can have an active role in the pollution episodes that China has been suffering in recent years. Thus this paper is within the scope of ACP and I would recommend its publication after addressing some concerns. The text needs some rewriting to make it clearer to the reader, especially the part reporting the TVOC measurements and the modeling of fluxes from those measurements.

Response: We thank Anonymous Reviewer#2 for his or her comments and appreciate the constructive criticisms which improve largely the presentations and interpretations in our manuscript. Based on the comments from the Reviewer #2, we have made corresponding revisions to the manuscript. Following are reviewer's comments and our responses

Specific comments:

P2L2: correct the number "R2=0014", there must be a decimal point missing.

Response: Corrected. Thanks!

P2L10: defining reactive BVOCs emitted by plants as "harmful gases" is not appropriate. Authors can argue that they have implications for atmospheric generation of pollutants such as ozone, but not that these gases are harmful.

Response: We agree with the reviewer's comment. In the revised manuscript, we have rewritten text as "they also contribute to air pollution through atmospheric chemistry"

P6L19: Reference to Guenther et al 2006, is it correct? If the MEGAN version was 2.1, should this reference be Guenther et al 2012? Otherwise, MEGAN version should be 2.0.

Response: The reference should be Guenther et al 2012. We have changed '2006' to '2012'. Thanks!

P9L1: Should L (oxidation rate) be replaced with C (isoprene concentration) in the text? Table S3 does not list L but C, and it is reasonable that L will actually vary with OH and O3 concentrations, which are also listed in this list.

Response: We agree with the reviewer's comment! The texts have been changed to ' the measured isoprene concentration (C)'.

P10L7: Please write the genus name Populus starting with capital letter. Was it only one species of poplars that were planted in the region? If so, please give the scientific name, otherwise list as Populus spp and refer to this trees in plural in the text.

Response: Thanks for the suggested changesïijĄWe have rewritten 'populus' to 'Populus spp' and referred to this trees in plural in the revised text (e.g., changing 'poplar' to 'poplars').

P10L10: Please list the variety of P. sylvestris or otherwise remove the word "var".

Response: We have removed 'var' following the reviewer's comment.

P13L13: The slope of -0.534 applies to northern China without including the TNRSF, according to Fig S5. Please clarify in the text.

Response: The text has been rewritten as '... the relatively strong increasing trend (Fig. 2) in the TNRSF (slope=0.881, R=0.579) has reversed the negative trend (slope=-0.533, R=0.224) of the total annual isoprene emissions in Northern China, which did not take the isoprene emissions in the TNRSF into consideration, to the positive trend (slope=0.347, R=0.118)...'.

P14L14: Did the authors do any statistical analysis to support the statement that the trends of isoprene emissions in the Central-North region are statistically significant whereas those from the other two regions are not?

Response: In the revised manuscript, we added p values for each trend in the three regions. As shown, the p value=0.002 for the isoprene emission trend in the Central-

North regionïijŇindicating statistically significant trend. Relatively weak significant trend was found in the Northwestern China region (p=0.012), and no statistically significant trend existed in the Northeastern China (p=0.484). These have been incorporated in the revised paper.

P17L8-P18L2: Please clarify this part of the text. If the surface of a model grid square is not completely covered by vegetation, wouldn't this imply that the calculated MEGAN fluxes do not compare so nicely with the estimates using Eq (1), mainly because the MEGAN fluxes calculated for these sites where TVOC measurements were performed would be higher (more vegetation coverage than the model grid square)?

Response: The reviewer raised a good point! To address the reviewer's question, we have extended discussions on potential reasons causing the difference between MEGAN modeled and TVOC measurements converted fluxes. Except for the reason the reviewer questioned, we also considered an additional cause: in the simplified Gaussian model (Supplementary) we choose the fetch $\Delta l$ =3km which is related directly to the magnitude of the converted emission fluxes which were subject to uncertainties. Nevertheless, overall the converted fluxes from the measured TVOC concentrations using the simplified Gaussian model are about the 2 fold of the modeled fluxes, suggesting the reasonable accuracy of the MEGAN model applied in the present investigation. These discussions have been incorporated into the revised manuscript.

P18L14: was vehicular exhaust the dominant source of atmospheric isoprene? Do the authors want to say that vehicular exhaust was the dominant source of atmospheric VOCs? Same for line 17 of this page.

Response: We have rephrased text in these two sentences. In the revised text, we made clear that ' the atmospheric isoprene during the wintertime was emitted mostly from vehicular exhaust', and 'the summertime isoprene was released from biogenic sources'.

P19L15: Correct Arneths to Arneth.

Response: Done! Thanks!

P20L10-13: Please clarify this sentence.

Response: Following the reviewer's comment, we have rephrased this sentence to make our point more clear.

P22L1: This sentence needs more information to make sense. As it currently reads, it may seem that 2007 was a bad year for the trees, but looking at Fig. 2, isoprene emissions are at or near the historical maximum. I suspect the authors have something else in mind that is not clear to me. What is the time span that the authors describe as showing a "considerable decline of forest coverage and isoprene emissions"?

Response: We thank the reviewer to point out this inconsistence. The forests collapse took place since 2007 rather than in 2007 (see Zhang, X., et al., 2015 in the Reference). We have replaced 'in' by 'since' in the revised paper. We further indicated that the mortality of trees since 2007 caused visible decline of the forest coverage and isoprene emissions in this region after 2007.

P22L7-8: the authors assume steady state of the mixed forest of Northeast China, regarding which variable? LAI? If so, have the authors checked whether the LAI information on Fig S6 agree with this assumption?

Response: The LAI data did show no trend in Northeast China. But in the revised manuscript we have deleted 'steady state'. Instead, we added new text, a new Fig. S7b which shows annual temperature averaged over Northeastern China, and corresponding discussions in the revised paper.

P27L10: Please list the year of publication (1996) and the complete list of authors.

Response: Done!

FigS6 (caption): LAT should be LAI?

Response: Yes, 'LAT' is 'LAI'. This error was corrected in the revise paper. Thanks!

---

## Referee Comment (RC3) · Anonymous Referee #3 · 15 Mar 2016

Long term trend of isoprene emission in the Three Northern Regions Shelter Forest (TNRSF) from 1982 to 2010 was evaluated, using a biogenic emission model for gases and aerosols (MEGAN). Isoprene emission flux has increased substantially in many places in the TNRSF due to the increase of trees and vegetation coverage, especially in the Central-North China region. The estimated isoprene emissions suggest that the TNRSF has altered the long-term emission trend in North China. I recommend its publication after addressing some questions. Please see the questions and comments bellow:

Specific comments:

P2L10: they also emit harmful gases into the air. By our understanding, these gases

are not harmful, please correct it, or cite references here. P6L12: MEGAN2.1 is primarily driven by biological and meteorological factors, including vegetation type with which the emission factors of BVOCs are assigned, air and leaf temperatures, light, leaf age and leaf area index (LAI), and soil moisture. Please introduce these data sources in the calculation for past three decades, the uncertainties of all these parameters used in the model for TNRSF, for example PAR, emission factor. P9L6: What are the sampling numbers at 8 sites in a field campaign? More introductions should be given for the measurements, such as VOC species and concentrations. P12L10: It's better to use mg m-2h-1 instead of micro-moles m-2 hr-1. P14L17 (and P22L14): These natural forests already reached the steady state, is there any evidence from botanical field? P15L13: No direct measurements of BVOCs emission data across the TNRSF have been ever reported before. This sentence should be corrected. Some measurements of BVOC emissions and concentrations in TNRSF region had been carried out, for example: Klinger L.F., Li Q.J., Guenther A. et al. 2002. Assessment of volatile organic compound emissions from ecosystems of China, J. Geophy. Res., 107(D21). Wang Z.H., Bai Y.H., Zhang S.Y., 2003. A biogenic volatile organic compounds emission inventory for Beijing. Atmospheric Environment. 37, 3771-3782. Bai J.H., Baker B., Liang B.S., Greenberg J., Guenther A., 2006. Isoprene and monoterpene emissions from an Inner Mongolia grassland. Atmospheric Environment. 40(30), 5753-5758. There should be more others that can be used for the evaluation.

P19L12: The increasing biogenic isoprene emissions can be attributed to the development of the TNRSF (i.e., LAI), how about the roles of other factors, PAR, temperature. My suggestion is to consider these parameters in all analysis, including P24L3. Where are figures Fig. S6a and b? P24L16: emission minus dry deposition? please make it clear.

---

## Author Comment (AC3) · 24 Mar 2016

Anonymous Reviewer #3

Long term trend of isoprene emission in the Three Northern Regions Shelter Forest (TNRSF) from 1982 to 2010 was evaluated, using a biogenic emission model for gases and aerosols (MEGAN). Isoprene emission flux has increased substantially in many places in the TNRSF due to the increase of trees and vegetation coverage, especially in the Central-North China region. The estimated isoprene emissions suggest that the TNRSF has altered the long-term emission trend in North China. I recommend its publication after addressing some questions. Please see the questions and comments bellow:

[Figure]

Response: We appreciate Anonymous Reviewer#3 for his or her comments and the constructive criticisms which help us to improve considerably our manuscript. Based on the comments from the Reviewer #3, we have made relevant revisions to the manuscript. Following are reviewer's comments and our responses.

Specific comments: P2L10: they also emit harmful gases into the air. By our understanding, these gases are not harmful, please correct it, or cite references here.

Response: We agree with the reviewer's comment! In the revised manuscript, we have rewritten text as "they also contribute to air pollution through atmospheric chemistry"

P6L12: MEGAN2.1 is primarily driven by biological and meteorological factors, including vegetation type with which the emission factors of BVOCs are assigned, air and leaf temperatures, light, leaf age and leaf area index (LAI), and soil moisture. Please introduce these data sources in the calculation for past three decades, the uncertainties of all these parameters used in the model for TNRSF, for example PAR, emission factor.

Response: Following the reviewer's comment, in the revised paper we have added text describing the data sources of meteorological data used in the modeling 30 years isoprene emissions, including a website and a new reference (Zhang et al., 2002). In the uncertainty analysis, we referred to Situ et al's work (2014) for the uncertainty analysis of the MEGAN model in which PAR and temperature were found to be most important environmental factors contributing to the uncertainties of the MEGAN model. We also further introduced PAR as one of the MEGAN model input parameters in our uncertainty analysis and rerun the Monte Carlo model. New results are presented in revised Fig. S1 and Table S2 of Supplementary materials.

P9L6: What are the sampling numbers at 8 sites in a field campaign? More introductions should be given for the measurements, such as VOC species and concentrations.

Response: The sampling frequency was set at 1 min. Since the GreyWolf VOC sensor

**[ACPD](ACPD)**
can only measure TVOC, the concentration of individual VOC species is not reported here. These have been mentioned in sections 2.4 and 3.4 in the previous and revised paper. Figure 8 illustrates sampled TVOC concentrations per minute.

P12L10: It's better to use mg m-2h-1 instead of micro-moles m-2 hr-1.

Response: Thanks for the suggestion! The use of this unit followed Guenther et al. (2012, see Reference list). We figured out that using this unit we could better compare and illustrate emission fluxes inside and outside the TNRSF. For instance, Figure 7 compares isoprene emission fluxes within the TNRSF, a natural forest in Northeastern China, and outside the TNRSF. Using micro-moles m-2 hr-1 the annual variation of the emission fluxes in these three regions can be nicely presented in the same panel of the figure. Whereas, the use of mg m-2 h-1 the annual fluxes outside the TNRSF cannot be shown in Fig. 7 because the fluxes become too small using this unit as compared with those within forests. Therefore, we prefer to use micro-moles m-2 hr-1.

P14L17 (and P22L14): These natural forests already reached the steady state, is there any evidence from botanical field?

Response: We don't have botanical data to show evidence of steady-state natural forest in Northeast China. Instead, we added two references which indicated that a natural forest could be assumed to be in a steady state. In the revised manuscript, we rewrote the text as " This implies that this natural forest was likely under a steady state from which the biogenic isoprene emissions were not altered on the decadal basis (Sanderson et al., 2003; Purves et al., 2004)".

P15L13: No direct measurements of BVOCs emission data across the TNRSF have been ever reported before. This sentence should be corrected. Some measurements of BVOC emissions and concentrations in TNRSF region had been carried out, for example: Klinger L.F., Li Q.J., Guenther A. et al. 2002. Assessment of volatile organic compound emissions from ecosystems of China, J. Geophy. Res., 107(D21). Wang Z.H., Bai Y.H., Zhang S.Y., 2003. A biogenic volatile organic compounds emission

inventory for Beijing. Atmospheric Environment. 37, 3771-3782. Bai J.H., Baker B., Liang B.S., Greenberg J., Guenther A., 2006. Isoprene and monoterpene emissions from an Inner Mongolia grassland. Atmospheric Environment. 40(30), 5753-5758. There should be more others that can be used for the evaluation.

Response: We thank the Reviewer#3 for letting us know these references. We did not cite some of these references because the field measurements reported in these references were not conducted in the TNRSF. e.g., Wang et al.'s field work was done in Beijing and Bai et al.'s study was focused on the grassland of Inner Mongolia. Nevertheless, following the reviewer's comment we have revised the text as "No extensive and direct measurements of BVOC emission across the TNRSF have been ever carried out. Several field campaigns have been conducted to measure BVOC emissions in Northern China but they were not typically designated for the TNRSF (Klinger et al., 2002; Wang et al., 2003)". These two references are also added to the Reference list.

P19L12: The increasing biogenic isoprene emissions can be attributed to the development of the TNRSF (i.e., LAI), how about the roles of other factors, PAR, temperature. My suggestion is to consider these parameters in all analysis, including P24L3. Where are figures Fig. S6a and b?

Response: We agree with Reviewer's comments! BVOC emissions do depend on other parameters and factors. We have shown that lower temperature in 2009 than 1982 in the Northeast China region of the TNRSF led to lower isoprene fluxes in 2009 in this region. In the revised manuscript, we have also added Fig. S7b which further shows decreasing trends of annually averaged temperature in the Northeast China region where LAI showed the incline (Fig. S6a) but BVOC emissions exhibited negative trends (Figs. 4 and 5). We suggested that declining temperatures in this region might cause the decreasing trend of isoprene emissions. Corresponding discussions have been incorporated in the Discussion section in the revised manuscript. We also wrote " Another environmental factor that may exert strong influence on the trend of isoprene emissions is solar radiation/PAR (Situ et al., 2014). Analogous to the response of

the BVOC emissions to temperature, increasing radiation could also enhance the isoprene emissions, or vice versa, particularly on daily or monthly basis." in the revised manuscript.

Fig. S6a and b are cited in Discussion section and presented in Supplementary materials.

P24L16: emission minus dry deposition? please make it clear.

Response: Thanks to the Reviewer to indicate this error. We have removed texts in the revised manuscript.
* * *

---

## Author Response (AR2)

**Response to the Editor and Reviewer#3 comments**

First of all, we would like to thank the Editor for his constructive comments and suggestions during the course of the revision of this paper, which improve greatly the interpretations and presentation of the manuscript. Following the Editor's suggestion, we have removed those discussions related to the ambient observation of TVOC and inverse modeling using box models. We also deleted the corresponding references, figures, and tables in main text and Supplement. Detailed responses to the Editor and Reviewer#3's comments are presented below.

REVIEWER COMMENT: L43: "they also contribute to air pollution through atmospheric chemistry"

EDITOR SUGGESTION: I suggest changing "they also contribute to" to "they also play a role in"

AUTHOR RESPPONSE: Done, thanks!

REVIEWER COMMENT:L64: "Efforts have been also made to measure and simulate BVOC emissions in China", my suggestion is more other studies should be introduced and cited here.

EDITOR SUGGESTION: No further revisions are needed.

REVIEWER COMMENT: L96: please introduce the coverage of the Three Northern Regions Shelter Forest (TNRSF) in the text and in the figure 1. Does it cover the Inner Mongolia and Beijing area? According to public references, the TNRSF covers Inner Mongolia and Beijing. Why the authors say differently?

EDITOR SUGGESTION: I assume the "green" area in Figure 1 is the TNRSF but you should make this clear. In that case it appears that Beijing and inner Mongolia are in the TNRSF but again you should clarify this. Also, you should make it clear that there are areas in the TNRSF zone but they are not (yet) forested areas.

AUTHOR RESPPONSE: Yes the TNRSF covers Beijing and Inner Mongolia. In Fig. 1 caption we have listed all provinces and two megacities, Beijing and Tianjin covered by the different regions of the TNRSF. As shown by Figure 1, both Beijing and Inner Mongolia are included in the TNRSF. To clarify this, in the revised paper we add text " **Figure 1** illustrates the TNRSF regions, including 11 provinces and two megacities, Beijing and Tianjin, as highlighted in the figure caption and marked in the figure." (line 98-100). In Fig. 1 caption, we added more details in the coverage of the TNRSF and stated that " many places in this part of the TNRSF, particularly in Gansu, Ningxia, and West Inner Mongolia, are not covered by forest but by shrubs".

REVIEWER COMMENT: L144: PAR was calculated from solar radiation provided by the Big-leaf dry deposition model. Are these PAR values compared with the ground observations at some sites for hourly and daily PAR? What are the calculating errors in TNRSF? Then, what uncertainties would be caused to BVOC emissions?

EDITOR SUGGESTION: No further revisions are needed.

REVIEWER COMMENT: L241: The GreyWolf VOC sensor can only measure TVOC, how authors examine and verify the release of BVOC species from the TNRSF? Based on measurements, the contributions of isoprene to BVOC emissions or to TVOC emissions are vegetation and season dependent, assuming the isoprene emission to be 50% of the TVOC would cause large errors.

EDITOR SUGGESTION: Change "this reactive biogenic VOC species" to "total biogenic VOC species"

AUTHOR RESPONSE: The paragraph with these texts have been deleted in the revised paper.

REVIEWER COMMENT:L263-264:"…suggesting increasing isoprene emissions associated with the expansion of the TNRSF in these regions". Is it the only factor for isoprene emission increasing? How about the roles of other factors, for example, PAR and temperature?

EDITOR SUGGESTION: This is an important point. At a minimum you should address it by including temperature and PAR trends in the figure.

AUTHOR RESPONSE: We did recognized the important role of temperature in isoprene emission. Since this study focused on long-term trend of isoprene emission, we have mentioned the influence of climate change, characterized by changes in mean temperature, on isoprene emission, e.g., lines 76, 89-91, 390-405, and Fig.S5. In the revised paper, following the Editor and Reviewer#3's comments, we replaced Fig. S7a by Fig. S5a which displays the differences of surface temperatures $T_{dif}$ between 1982 and 2012 across the TNRSF (now Fig. S5a), instead of $T_{dif}$ in the Northeast China region only as presented by Fig. S7 in the previous Supplement of the manuscript. Figure 5b presents 30 years trend of SATs from 1982 to 2010 across the Northeast China region of the TNRSF which was used to explain negative trend of isoprene emissions in this part of the TNRSF (Fig. 4b). Corresponding discussions are presented in Discussion section (Section 4, line 386-405). In Section 3.2, we also added a paragraph " As aforementioned in Introduction, in addition to forest expansion, biogenic isoprene emissions are also associated with climate change via changes in mean temperature (Sanderson et al., 2003) and PAR (Guenther et al., 2006, 2012; Situ et al., 2014). Since the influence of climate change on BVOC is beyond scope of this article, we shall not assess detailed associations between climate change (mean temperature) and isoprene emissions from the TNRSF.

Nevertheless, in Section 4, we shall discuss briefly the potential influence of the changes in annual mean air temperature and PAR on long-term trends of biogenic isoprene emissions in the Northeast China region of the TNRSF."

In new Fig. S6, we presented the linear trends of PAR from 1982-2010. We added a new paragraph (line 406-429) to discuss and interpret the trend of PAR. Results suggested that PAR was unlikely to overwhelm the long-term trend of isoprene emissions.

REVIEWER COMMENT:L587: "Since VOCs are major precursor" should be changed to "Since BVOCs are major precursors".

EDITOR SUGGESTION: You should make this change

AUTHOR RESPONSE: Done!

**Marked up manuscript**

[revised manuscript text omitted]